

# Identification and functional analysis of PIN family genes in *Gossypium barbadense*

Yilei Long[1,*], Quanjia Chen[1,*], Yanying Qu[1], Pengfei Liu[1], Yang Jiao[1], Yongsheng Cai[1], Xiaojuan Deng[1] and Kai Zheng[1,2,3]

[1] College of Agronomy, Xinjiang Agricultural University, Urumqi, Xinjiang, China
[2] Hainan Yazhou Bay Seed Laboratory, Sanya, Hainan, China
[3] Postdoctoral Research Station, Xinjiang Agricultural University, Urumqi, Xinjiang, China
* These authors contributed equally to this work.

Corresponding author
Kai Zheng, zhengkai555@126.com

## ABSTRACT

**Background:** PIN proteins are an important class of auxin polar transport proteins that play an important regulatory role in plant growth and development. However, their characteristics and functions have not been identified in *Gossypium barbadense*.
**Methods:** PIN family genes were identified in the cotton species *G. barbadense*, *Gossypium hirsutum*, *Gossypium raimondii*, and *Gossypium arboreum*, and detailed bioinformatics analyses were conducted to explore the roles of these genes in *G. barbadense* using transcriptome data and quantitative reverse-transcription polymerase chain reaction (qRT-PCR) technology. Functional verification of the genes was performed using virus-induced gene silencing (VIGS) technology.
**Results:** A total of 138 PIN family genes were identified in the four cotton species; the genes were divided into seven subgroups. *GbPIN* gene family members were widely distributed on 20 different chromosomes, and most had repeated duplication events. Transcriptome analysis showed that some genes had differential expression patterns in different stages of fiber development. According to 'PimaS-7' and '5917' transcript component association analysis, the transcription of five genes was directly related to endogenous auxin content in cotton fibers. qRT-PCR analysis showed that the *GbPIN7* gene was routinely expressed during fiber development, and there were significant differences among materials. Transient silencing of the *GbPIN7* gene by VIGS led to significantly higher cotton plant growth rates and significantly lower endogenous auxin content in leaves and stems. This study provides comprehensive analyses of the roles of PIN family genes in *G. barbadense* and their expression during cotton fiber development. Our results will form a basis for further PIN auxin transporter research.

## INTRODUCTION

Auxins play an important role in plant root development (*Dong et al., 2018*), apical dominance (*Wickson & Thimann, 2010*), embryogenesis (*Hellmann, 2014*), vascular differentiation (*Dubrovsky et al., 2008*), tropism (*Rakusová, Fendrych & Friml, 2015*), and responses to internal (*Mashiguchi et al., 2011*) and external stimuli. They are primarily synthesized in the terminal and lateral buds and root tips, and then migrate to various

plant parts *via* polar or non-polar transport (*Zheng et al., 2013*; *Yang et al., 2015*). Current research on auxin polar transporters focuses on three categories: P-glycoprotein (MDR/PGP/ABCB) efflux/conditional transporters (*Grones & Friml, 2015*), class II auxin antibody (AUX1/LAX) intrafluidic vectors (*Péret et al., 2012*), and plant-specific PIN-FORMED (PIN) efflux carriers (*Yu, 2009*). PIN proteins are important auxin polar transporters that play important regulatory roles in plant growth and development (*Zhang et al., 2019*; *Gan et al., 2019*; *Qu et al., 2020*). The polar subcellular localization of PIN efflux proteins on the plasma membrane determines the directional flow of auxin (*Barbez et al., 2012*).

The *Arabidopsis* genome encodes eight PIN proteins, which are divided into two categories according to their subcellular locations and functions. PIN type I proteins include *PIN1*, *PIN2*, *PIN3*, *PIN4*, and *PIN7*, which are located on the cytoplasmic membrane and participate in root development and various abiotic stress responses (*Gälweiler et al., 1998*; *Feraru et al., 2012*; *Friml, 2013*), and PIN type II proteins include *PIN5*, *PIN6*, and *PIN8*, which are localized to the endoplasmic reticulum and are involved in auxin homeostasis (*Mravec et al., 2009*; *Ding et al., 2012*). In addition, seven PIN-LIKE (PILS) proteins have a topology similar to that of PIN proteins and contribute to the stabilization of intracellular auxin (*Feraru et al., 2012*; *Yue et al., 2015*). Thus, PIN and PILS genes constitute the auxin efflux PIN gene family, which is responsible for the transport of auxin from the point of biosynthesis to the site of action, and establishes a concentration gradient to achieve its regulatory function (*Feraru et al., 2019*).

With the publication of genome-wide data for various plants, the PIN gene family has been identified in increasing numbers of plant species. *Xu et al. (2005)* identified 12 PIN family genes in rice; an experiment in which light was applied to one side of growing rice roots showed that the expression levels of four genes, *OsPIN1a*, *OsPIN1b*, *OsPIN1c*, and *OsPIN1d*, were significantly increased on the light side (*Liang et al., 2017*), suggesting that the *OsPIN* gene family plays an important role in root development. Fourteen PIN genes have been identified in the maize genome, among which 13 transcripts have been detected in grains, indicating that the *ZmPIN* gene plays an important role in inflorescence development in maize (*Li & Meng, 2019*). Overexpression of *OsPIN1a* in rice and *ZmPIN1a* in maize promotes root tillering and regulates root growth and tissue differentiation (*Carraro et al., 2006*; *Xu et al., 2014*). High expression of PttPIN1 in poplar, LaPIN1 in white lupine and SlPIN1 in tomato promote the growth of vascular cambium and hypocotyls, as well as the development of vegetative organs and young fruit (*Schrader et al., 2003*; *Oliveros-Valenzuela et al., 2007*; *Nishio et al., 2010*; *Pattison & Catalá, 2012*). *Arabidopsis AtPIN1* and *AtPIN2* gene deletion mutations have been found to cause inflorescence deformity (*Gälweiler et al., 1998*) and increase the density of lateral root hairs (*Lin et al., 2012*). Downregulated expression of the tomato gene *SiPIN8* causes pollen abortion (*Gan et al., 2019*).

*PIN* genes have been widely researched in cotton. Detailed analyses of the *Gossypium hirsutum* genome (*Zhang et al., 2017a*; *He et al., 2017*) have identified 17 PIN auxin efflux carriers, among which *PIN1–3* and *PIN2* in the At subgenome are highly expressed in roots (*He et al., 2017*). PIN genes exhibit different induction effects under different abiotic

**Table 1 Fiber quality data for two sea-island cotton materials from 2018 to 2020.**

| Name | Fibre length/mm | Fibre strength/cN/tex | Micronaire | Fibre elongation/% | Fibre uniformity/% | Fibre maturity | Spinning consistence index |
|------|------|------|------|------|------|------|------|
| PimaS-7 | 34.07 ± 1.85 | 37.02 ± 0.41 | 4.77 ± 2.45 | 4.33 ± 0.73 | 86.67 ± 2.16 | 0.88 ± 0.02 | 188.5 ± 7.38 |
| 5917 | 35.19 ± 2.38 | 49.61 ± 2.85** | 4.92 ± 0.86 | 4.48 ± 0.48 | 87.29 ± 3.8 | 0.88 ± 0.01 | 214.73 ± 37.53 |

Note:
** Indicates that this trait is significantly different between the two materials. $P < 0.01$.

stresses (Zeng et al., 2019). Elements corresponding to auxin and salicylic acid are found in the promoter regions of most PIN genes (Zhang et al., 2017a). However, the roles of PIN genes have not yet been studied in *Gossypium barbadense*. QTL analysis has shown the involvement of PIN genes in *G. barbadense* fiber strength (Li et al., 2017b); however, their number and identity remain unknown. The completion of *G. barbadense* genome sequencing has facilitated the identification of polar auxin transport genes (Wang et al., 2019). Therefore, in this study, we conducted comprehensive bioinformatics analyses to identify PIN genes in *G. barbadense*. Data on PIN family gene expression collected during fiber development were analyzed to explore the relationship between endogenous auxin content and gene expression, and virus-induced gene silencing (VIGS) of key genes was performed to examine the role of auxin polar transport carrier proteins in plant growth and development.

# MATERIALS AND METHODS

## Planting and sampling of cotton material

We used PimaS-7 (low fiber strength) and 5917 (high fiber strength) *G. barbadense* resource materials that had been preserved by the Key Laboratory of Crop Genetic Improvement and Germplasm Innovation, College of Agriculture, Xinjiang Agricultural University, Urumqi, China. The fiber characteristics of the samples are provided in Table 1.

The test materials were planted in an experimental field of the Xinjiang Academy of Agricultural Sciences, Urumqi, China. Cotton fiber samples were collected at 0, 5, 10, 15, 20, 25, 30, and 35 days post-anthesis (DPA), immersed in liquid nitrogen, and refrigerated at −80 °C until use.

## Identification and physicochemical properties of PIN genes

We downloaded PIN genomic data for *G. hirsutum* TM-1 (ZJU-AD1_v2.1_a1.0) (Hu et al., 2019), *G. barbadense* Pima3–79 (HAU_v2.0) (Wang et al., 2019), *Gossypium arboreum* (CRI-A2_v1.0) (Du et al., 2018), and *Gossypium raimondii* (JGI_221_v2.1) (Paterson et al., 2012) from the Cottongen database (http://www.cottongen.org). PIN family accession numbers were cross-referenced with Pfam database (https://pfam.xfam.org/) and the associated Markov model files were downloaded. A local database was constructed using HMMER 3.3.2 software to retrieve the initial files, and the amino acid sequences were reconstructed and loaded into the Pfam database for searching (Ji et al., 2022). All amino acid sequences that did not contain the desired conserved structure were

deleted from the original sequence, and a complete PIN family member result was obtained. The chromosomal locations and DNA lengths were verified using the cottonfgd database (https://cottonfgd.net/) and ExPASy program (https://web.expasy.org/protparam/) was used to predict the length, molecular weight (MW), isoelectric point (pI), instability coefficient, and hydropathic index of the corresponding proteins.

## Phylogenetic analysis and gene chromosome mapping and replication

The amino acid sequences of PIN genes were loaded into the MEGA-X software, and the neighbor-joining method was used to construct an evolutionary tree, which was visualized using the iTOL online tool (https://itol.embl.de/). PIN gene conserved domain positions were determined using the National Centers for Biotechnology Information (NCBI) browser (https://www.ncbi.nlm.nih.gov/Structure/bwrpsb/bwrpsb.cgi) and the Clustal Omega tool (https://www.ebi.ac.uk/tools/msa/clustalo/); Jalview software was used for sequence alignment and to visualize the results.

The TBtools online toolkit was used to obtain chromosome density information from genome annotations, and position the PIN genes on chromosomes. Amino acid sequences of PIN genes in the four cultivated cotton species were aggregated and collinear scanning was performed using MCScanX (*Wang et al., 2012*).

## Gene structure, motif, and promoter cis-element analysis

The amino acid sequences of PIN family members were loaded into the MEME tool (https://meme-suite.org/meme/tools/meme); the first 2,000 bp of the start codons of PIN genes were loaded into the PlantCARE database (http://bioinformatics.psb.ugent.be/webtools/plantcare/html/) to predict the motif structure and promoter elements of the PIN genes.

## Transcriptome analysis of PIN genes and determination of endogenous auxin content

The steps for the transcriptome analysis were as follows. Download transcriptome Hai7124 and TM-1 data at NCBI (https://www.ncbi.nlm.nih.gov/), then remove PimaS-7 and 5917 transcripts in the laboratory Group data (*Hu et al., 2019*). Using the *G. hirsutum* and *G. barbadense* genomes as reference genomes, conduct sequence alignment with HISAT2 (*Kim, Langmead & Salzberg, 2015*), then use feature Counts (*Liao, Smyth & Shi, 2014*) to obtain the raw count of each gene in each sample. Finally, import the data into R and use edgeR (*Robinson, McCarthy & Smyth, 2010*) for expression analysis. The method for determining endogenous auxin (IAA) in each period of fiber development adhered to the instruction manual of the plant auxin (IAA) enzyme-linked immunosorbent assay kit for fiber development, and this method was used to measure 0–35 DPA endogenous growth of PimaS-7 and 5917 (Fig. S1). SPSS was used to analyze the correlation between the endogenous auxin content and fragments per kilobase of exon model per million mapped fragments (FPKM) value of the PIN family gene at each stage of fiber development.
## RNA extraction and quantitative reverse-transcription polymerase chain reaction (qRT-PCR) analysis

Total RNA was extracted from the collected samples using a polysaccharide polyphenol plant extraction kit provided by Beijing Tiagen Reagent (Beijing, China), according to the manufacturer's instructions. The ABM reverse transcription kit was used to reverse convert the extracted RNA into first-strand cDNA, and the reaction was conducted according to the manufacturer's instructions. qRT-PCR analysis was performed as follows. The primer GB-UBQ7 was designed using the cotton *UBQ7* gene as the internal reference gene, and the relative expression levels of the target gene were calculated using the $2^{-\Delta\Delta Ct}$ method. The fluorescence quantitative reaction system consisted of 10 μL EVAGreen Express 2× qPCR, 2 μL cDNA template, 1 μL each of forward and reverse primers (10 μmol/L), and water to a final volume of 20 μL. The reaction conditions were 40 cycles of 94 °C for 30 s, 94 °C for 5 s, and 60 °C for 30 s.

## Transient silencing analysis of the *GbPIN7* gene

A cotton gene silencing system was established based on the tobacco rattle virus (TRV) vector (*Liu & Page, 2008*). The 320 bp specific nucleic acid fragment of the *GbPIN7* gene was subcloned and loaded into the target site of a pTRV2 vector. *Agrobacterium* containing pTRV1, pTRV2-GbPIN7, and pTRV2-CLA was mass cultured, followed by 1:1 configuration of the transformed bacteria. Working solutions of the experimental group (pTRV1+pTRV2::GbPIN7), positive control (pTRV1+pTRV2::CLA), and negative control (pTRV1+pTRV2 empty) were each injected into PimaS-7 and 5917 plants that had not yet grown true leaves. After inoculation, the plants were grown at a constant temperature of 25 °C in an artificial culture room under a 16 h/8 h light/dark cycle. After 2 weeks, the albino phenotype was observed and photographed. Leaves were collected from the negative control and experimental groups for qRT-PCR analysis.

# RESULTS

## Evolutionary analysis of GbPIN family members with respect to physical/chemical properties and chromosomal localization

A total of 138 PIN family genes were identified among the four major cotton genera, of which 45 were identified in *G. barbadense*, 46 in *G. hirsutum*, 24 in *G. raimondii*, and 24 in *G. arboreum*. The resulting phylogenetic tree (Fig. 1) had three primary branches; the PIN and PILs genes were clearly distinguished. Analysis of the conserved domains showed that the 138 genes were divided into seven subgroups: PIN1, PIN2/PIN3/PIN4, PIN5/PIN6/PIN8, and PILS1/PILS3 (Fig. 2). The PILS5, PILS6, and PILS2 results further refined the subfamily positions of each gene; the phylogenetic branches and conservative structure analysis results matched completely, indicating the reliability of the classification results.

The PIN family genes of *G. barbadense*, *G. hirsutum*, *G. arboretum*, and *G. raimondii* were distributed on 20, 21, 10, and 11 different chromosomes, respectively (Fig. 3).
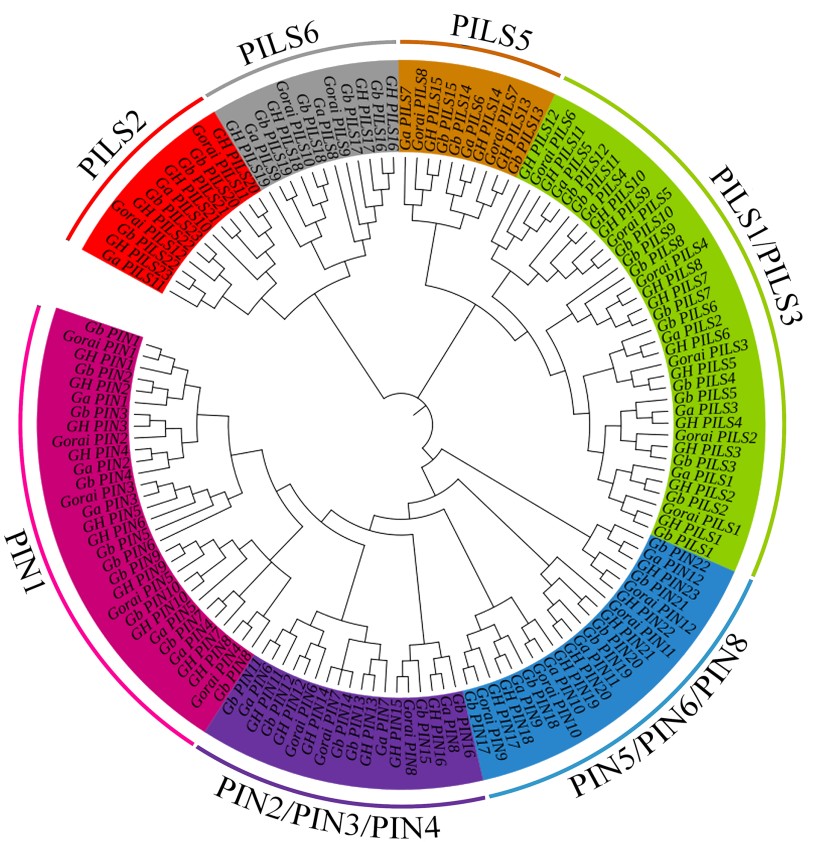

**Figure 1 Phylogenetic tree of PIN-FORMED (PIN) family genes in *Gossypium barbadense* (Gb), *Gossypium hirsutum* (Gh), *Gossypium arboreum* (Ga), and *Gossypium raimondii* (Gorai).** PIN family gene subgroups include PIN1, PIN2/PIN3/PIN4, PIN5/PIN6/PIN8, PILS1/PILS3, PILS5, PILS6, PILS2.

The GbPILS9, GbPILS14, and GbPIN4 genes in *G. barbadense* were identified as the extrachromosomal Scaffold gene structure, similar to the *G. arboreum* GaPILS7 gene. These results may have been caused by our gene assembly and annotation methods.

A total of 23 genes were identified using descriptive statistics for 45 sea-island cotton genes; these 45 PIN family genes are widely distributed on 20 different chromosomes; they are absent from only six chromosomes (A04, A06, A08, D06, D08, and D13). The lengths of the DNA sequences varied from 767 to 13,644 bp, and those of the coding sequences varied from 576 to 1,938 bp.

The amino acid sequence lengths ranged from 191 to 645; the pI values ranged from 4.85 to 9.65; protein molecular weights were distributed from 20.97 to 70.48 kDa; the hydropathic index ranged from 0.21 to 0.76, indicating that all members were hydrophobins. Among the 45 PIN family genes, 16 were unstable proteins, and 29 were stable structures. All members of the three PIN subgroups were localized on the cell membrane, whereas all members of the four PILS subgroups were localized outside the cell (Table 2).

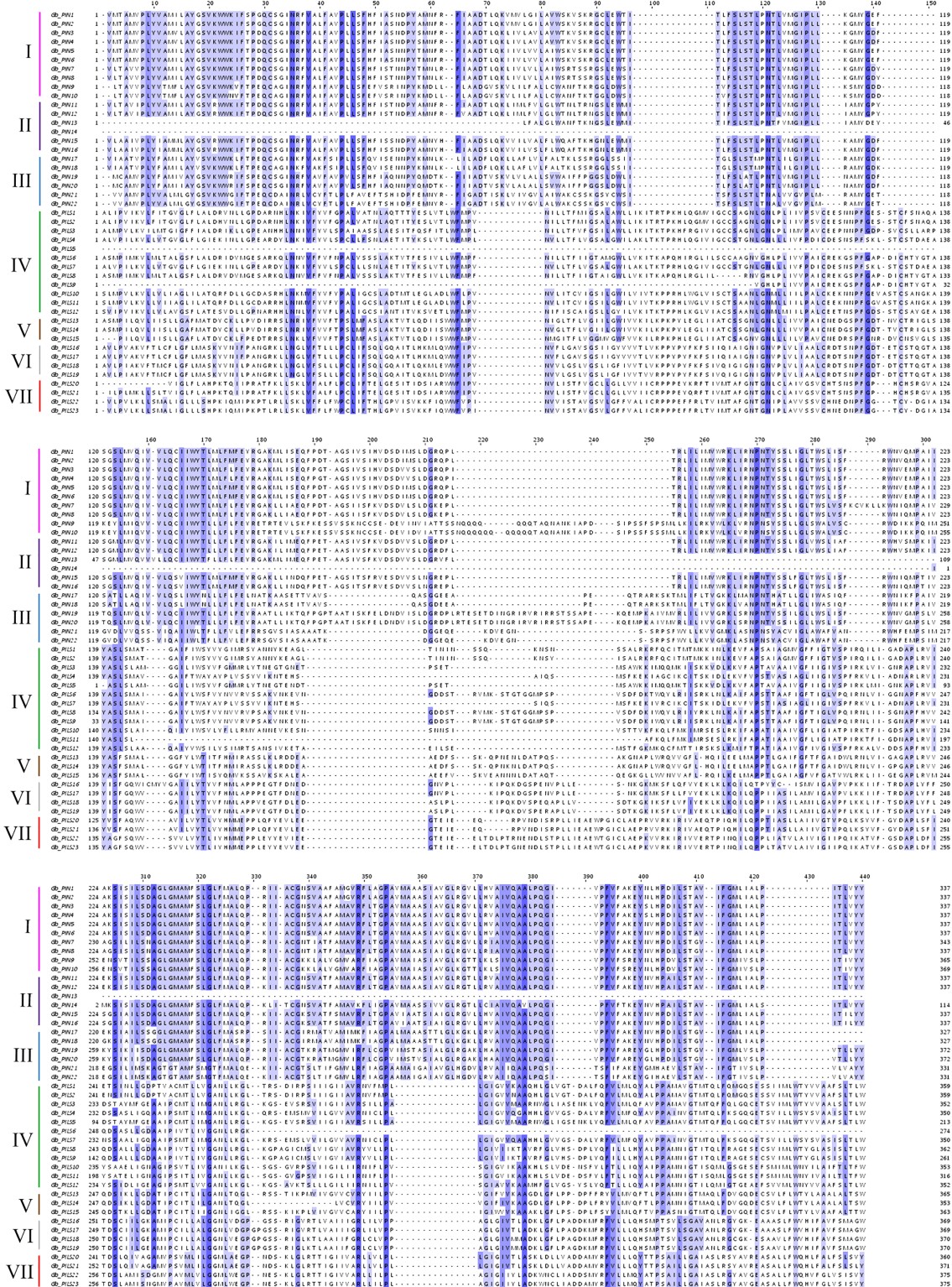

**Figure 2 Multiple sequence alignment of the conserved domains of members of the cotton GbPIN gene family.** Darker blue indicates higher conservation of amino acid residues. I–VII indicate different subfamilies, as follows: I, PIN1; II, PIN2/PIN3/PIN4; III, PIN5/PIN6/PIN8; IV, PILS1/PILS3; V, PILS5; VI, PILS6; VII, PILS2.

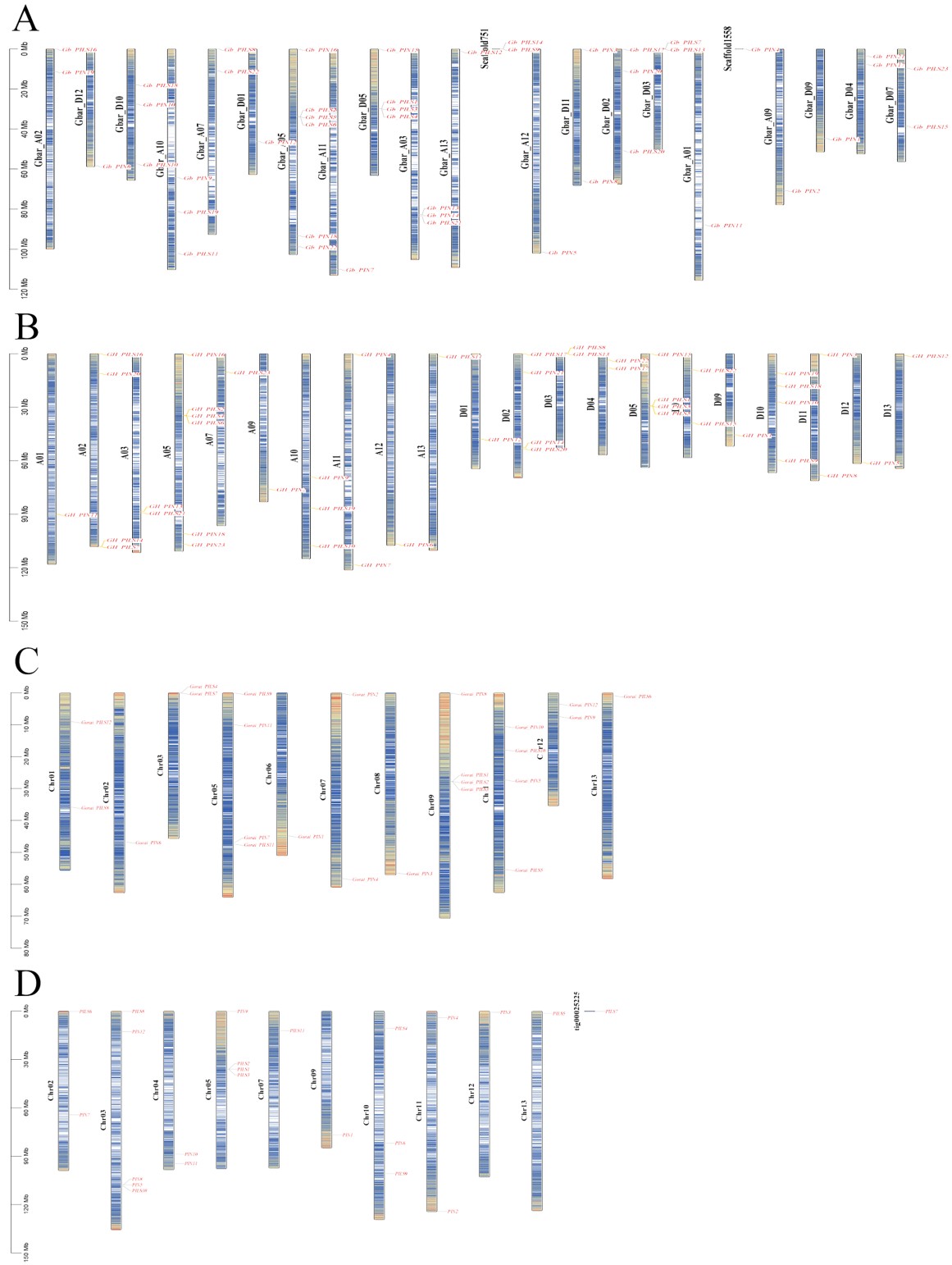

**Figure 3 Chromosome map of PIN family genes in four cultivated cotton species.** (A) *G. barbadense*, (B) *G. hirsutum*, (C) *G. arboreum*, and (D) *G. raimondii*.

**Table 2 Information on PIN family genes in *Gossypium barbadense*.**

| Number | Gene name | Gene ID | Chromosomal position | DNA length/bp | Open reading frame/bp | Protein length/bp | Theoretical isoelectric point (pI) | Stability factor | Is stable | Fat index | Hydrophilic index | Relative molecular weight (r)/kDa | Subcellular localization | Transmembrane structure |
|---|---|---|---|---|---|---|---|---|---|---|---|---|---|---|
| 1 | Gb_PIN1 | Gbar_D09G018320 | D09:44757059–44760734 | 3,676 | 1,275 | 424 | 8.92 | 36.1 | Yes | 87.88 | 0.07 | 66.09035 | Plasma membrane | exist |
| 2 | Gb_PIN2 | Gbar_A09G018560 | A09:70658205–70662107 | 3,903 | 1,818 | 605 | 8.92 | 35.88 | Yes | 87.88 | 0.08 | 66.04531 | Plasma membrane | exist |
| 3 | Gb_PIN3 | Gbar_D11G000270 | D11:234611–237723 | 3,113 | 1,854 | 617 | 9.11 | 36.5 | Yes | 88.69 | 0.10 | 66.73202 | Plasma membrane | exist |
| 4 | Gb_PIN4 | Gbar_D11G035200 | Scaffold1558:34946–38212 | 3,267 | 1,845 | 614 | 9.1 | 34.99 | Yes | 88.96 | 0.10 | 66.63696 | Plasma membrane | exist |
| 5 | Gb_PIN5 | Gbar_A12G028150 | A12:101432707–101436767 | 4,061 | 1,815 | 604 | 8.91 | 34.94 | Yes | 90.07 | 0.11 | 66.26874 | Plasma membrane | exist |
| 6 | Gb_PIN6 | Gbar_D12G028070 | D12:58395131–58399101 | 3,971 | 1,815 | 604 | 8.91 | 35.14 | Yes | 90.07 | 0.12 | 66.2871 | Plasma membrane | exist |
| 7 | Gb_PIN7 | Gbar_A11G032930 | A11:110063214–110067010 | 3,797 | 1,812 | 603 | 9.15 | 32.7 | Yes | 98.62 | 0.21 | 65.50486 | Plasma membrane | exist |
| 8 | Gb_PIN8 | Gbar_D11G033710 | D11:66000823–66004516 | 3,694 | 1,758 | 585 | 9.07 | 33.2 | Yes | 98.02 | 0.22 | 63.2011 | Plasma membrane | exist |
| 9 | Gb_PIN9 | Gbar_A10G013670 | A10:64236374–64238847 | 2,474 | 1,356 | 451 | 8.93 | 31.23 | Yes | 107.41 | 0.39 | 49.87675 | Plasma membrane | exist |
| 10 | Gb_PIN10 | Gbar_D10G014630 | D10:27566270–27568733 | 2,464 | 1,368 | 455 | 8.53 | 32.97 | Yes | 107.32 | 0.38 | 50.36816 | Plasma membrane | exist |
| 11 | Gb_PIN11 | Gbar_A01G014830 | A01:87873685–87878285 | 4,601 | 1,926 | 641 | 7.67 | 40.74 | No | 91 | 0.11 | 70.01387 | Plasma membrane | exist |
| 12 | Gb_PIN12 | Gbar_D01G015860 | D01:46277652–46291295 | 13,644 | 1,938 | 645 | 8.18 | 41.21 | No | 90.42 | 0.08 | 70.48137 | Plasma membrane | exist |
| 13 | Gb_PIN13 | Gbar_A03G013930 | A03:82788534–82789300 | 767 | 576 | 191 | 5.03 | 43.52 | No | 93.4 | 0.21 | 21.46077 | Plasma membrane | exist |
| 14 | Gb_PIN14 | Gbar_A03G013940 | A03:82789481–82790661 | 1,181 | 600 | 199 | 6.29 | 29.26 | Yes | 114.72 | 0.55 | 20.97977 | Plasma membrane | exist |
| 15 | Gb_PIN15 | Gbar_D05G000160 | D05:215682–218303 | 2,622 | 1,896 | 631 | 9.1 | 44.68 | No | 91.35 | 0.16 | 68.52537 | Plasma membrane | exist |
| 16 | Gb_PIN16 | Gbar_A05G000120 | A05:155712–158307 | 2,596 | 1,884 | 627 | 9.12 | 44.9 | No | 92.09 | 0.16 | 68.4013 | Plasma membrane | exist |
| 17 | Gb_PIN17 | Gbar_D04G005310 | D04:7699476–7701598 | 2,123 | 1,077 | 358 | 9.65 | 33.62 | Yes | 124.33 | 0.73 | 38.72448 | Plasma membrane | exist |
| 18 | Gb_PIN18 | Gbar_A05G036890 | A05:93325397–93327581 | 2,185 | 1,077 | 358 | 9.65 | 32.86 | Yes | 126.79 | 0.76 | 38.66042 | Plasma membrane | exist |
| 19 | Gb_PIN19 | Gbar_A02G006800 | A02:11379370–11383561 | 4,192 | 1,644 | 547 | 8.98 | 33.37 | Yes | 103.02 | 0.36 | 59.79408 | Plasma membrane | exist |
| 20 | Gb_PIN20 | Gbar_D02G007740 | D02:10933954–10937800 | 3,847 | 1,644 | 547 | 8.89 | 32.57 | Yes | 102.49 | 0.35 | 59.87015 | Plasma membrane | exist |
| 21 | Gb_PIN21 | Gbar_D04G002630 | D04:3460788–3462831 | 2,044 | 1,068 | 355 | 5.84 | 33.76 | Yes | 109.04 | 0.71 | 38.46028 | Plasma membrane | exist |

(Continued)

| Number | Gene name | Gene ID | Chromosomal position | DNA length/bp | Open reading frame/bp | Protein length/bp | Theoretical isoelectric point (pI) | Stability factor | Is stable | Fat index | Hydrophilic index | Relative molecular weight (r)/kDa | Subcellular localization | Transmembrane structure |
|---|---|---|---|---|---|---|---|---|---|---|---|---|---|---|
| 22 | Gb_PIN22 | Gbar_A05G039360 | A05:98774643–98776473 | 1,831 | 1,068 | 355 | 5.84 | 32.93 | Yes | 110.14 | 0.72 | 38.44224 | Plasma membrane | exist |
| 23 | Gb_PILS1 | Gbar_D05G029520 | D05:29762521–29766286 | 3,766 | 1,254 | 417 | 8.11 | 38.61 | Yes | 113.88 | 0.55 | 45.51267 | Extracellular | exist |
| 24 | Gb_PILS2 | Gbar_A05G028660 | A05:33799395–33803368 | 3,974 | 1,254 | 417 | 6.94 | 38.19 | Yes | 115.52 | 0.58 | 45.56574 | Extracellular | exist |
| 25 | Gb_PILS3 | Gbar_D05G029530 | D05:29775699–29781500 | 5,802 | 1,209 | 402 | 9.47 | 38.7 | Yes | 120.82 | 0.79 | 43.66251 | Extracellular | exist |
| 26 | Gb_PILS4 | Gbar_D05G029540 | D05:29799741–29803605 | 3,865 | 1,242 | 413 | 6.88 | 43.08 | No | 123.17 | 0.65 | 44.29533 | Extracellular | exist |
| 27 | Gb_PILS5 | Gbar_A05G028670 | A05:33810678–33812798 | 2,121 | 843 | 280 | 9.19 | 37.6 | Yes | 107.29 | 0.61 | 30.36924 | Extracellular | exist |
| 28 | Gb_PILS6 | Gbar_A05G028680 | A05:33889084–33892520 | 3,437 | 1,242 | 413 | 9.39 | 37.67 | Yes | 111.58 | 0.46 | 41.31478 | Extracellular | exist |
| 29 | Gb_PILS7 | Gbar_D03G000040 | D03:48069–55775 | 7,707 | 1,143 | 380 | 8.29 | 42.41 | No | 125.52 | 0.66 | 44.4156 | Extracellular | exist |
| 30 | Gb_PILS8 | Gbar_A07G000030 | A07:22415–26932 | 4,518 | 1,260 | 419 | 9.48 | 40.78 | No | 123.05 | 0.67 | 45.62937 | Extracellular | exist |
| 31 | Gb_PILS9 | Gbar_A02G019170 | Scaffold751:45216–48940 | 3,725 | 933 | 310 | 9.16 | 39.96 | Yes | 117.29 | 0.55 | 33.37739 | Extracellular | exist |
| 32 | Gb_PILS10 | Gbar_D10G021870 | D10:57531335–57535566 | 4,232 | 1,200 | 399 | 9.01 | 40.28 | No | 131.73 | 0.79 | 43.5741 | Extracellular | exist |
| 33 | Gb_PILS11 | Gbar_A10G021620 | A10:102044538–102049540 | 5,003 | 1,005 | 334 | 8.89 | 42.88 | No | 137.84 | 0.97 | 36.24069 | Extracellular | exist |
| 34 | Gb_PILS12 | Gbar_A13G001280 | A13:1373215–1377236 | 4,022 | 1,200 | 399 | 6.59 | 42.66 | No | 123.66 | 0.76 | 43.039 | Extracellular | exist |
| 35 | Gb_PILS13 | Gbar_D03G000070 | D03:85140–88187 | 3,048 | 1,224 | 407 | 5.51 | 44.27 | No | 119.75 | 0.67 | 44.51579 | Extracellular | exist |
| 36 | Gb_PILS14 | Gbar_A02G019140 | Scaffold751:20919–23268 | 2,350 | 1,188 | 395 | 5.49 | 42.55 | No | 121.19 | 0.67 | 43.27938 | Extracellular | exist |
| 37 | Gb_PILS15 | Gbar_D07G018990 | D07:38632211–38635100 | 2,890 | 1,263 | 420 | 8.09 | 41.11 | No | 114.88 | 0.57 | 45.95341 | Extracellular | exist |
| 38 | Gb_PILS16 | Gbar_A02G000040 | A02:9455–11643 | 2,189 | 1,272 | 423 | 9.06 | 37.75 | Yes | 117.8 | 0.66 | 45.79152 | Extracellular | exist |
| 39 | Gb_PILS17 | Gbar_D02G000160 | D02:121499–124181 | 2,683 | 1,272 | 423 | 9.26 | 36.05 | Yes | 121.96 | 0.71 | 45.48327 | Extracellular | exist |
| 40 | Gb_PILS18 | Gbar_D10G011810 | D10:17869988–17873330 | 3,343 | 1,233 | 410 | 8.33 | 33.85 | Yes | 126.27 | 0.79 | 44.15691 | Extracellular | exist |
| 41 | Gb_PILS19 | Gbar_A10G016340 | A10:81201694–81205175 | 3,482 | 1,233 | 410 | 8.54 | 35.54 | Yes | 126.02 | 0.78 | 44.13586 | Extracellular | exist |
| 42 | Gb_PILS20 | Gbar_D02G015690 | D02:50956897–50960579 | 3,683 | 1,356 | 451 | 6.46 | 42.54 | No | 123.64 | 0.58 | 49.80192 | Extracellular | exist |
| 43 | Gb_PILS21 | Gbar_A03G014140 | A03:83557884–83560152 | 2,269 | 1,356 | 451 | 6.26 | 42.17 | No | 122.99 | 0.58 | 49.83393 | Extracellular | exist |
| 44 | Gb_PILS22 | Gbar_A07G008090 | A07:10999462–11002034 | 2,573 | 1,353 | 450 | 4.98 | 38.34 | Yes | 122.56 | 0.59 | 49.96908 | Extracellular | exist |
| 45 | Gb_PILS23 | Gbar_D07G008460 | D07:9570357–9572193 | 1,837 | 1,440 | 479 | 4.85 | 36.84 | Yes | 118.79 | 0.55 | 53.19263 | Extracellular | exist |

## Gene structure and promoter analysis of PIN family genes in sea-island cotton

The gene structure analysis results showed that among the 45 PIN family genes, all members of the three PIN subfamilies except *GbPIN13* and *GbPIN14* contained six motif structures, three of which were at the beginning of the sequence (Fig. 4B). The *GbPIN13* gene contained only motif2 and motif6, and *GbPIN14* contained only motif3 and motif4. Fewer motif structures were found in each member sequence of the four PILS subgroups, and the most conserved motif1 structure was absent from 10 genes including *GbPILS3* and *GbPILS5*. This indicates that sequence conservation of PIN subfamily genes was significantly higher than that of PILS subfamily genes. In terms of the sequence structure of gene family members, only *GbPILS21*, *GbPILS22*, and *GbPILS23* had no introns, whereas most genes had large numbers of introns. PIN subfamily genes had significantly fewer introns. Among the four PILS subfamilies, nine genes, including *GbPILS11*, *GbPILS8*, and *GbPILS14*, did not have untranslated regions (UTRs). Two genes (*GbPILS7* and *GbPILS9*) contained only 3′ UTRs, and the remaining 34 genes had two UTRs (Fig. 4C).

To analyze the transcriptional regulation and potential functions of the cotton PIN family genes further, we predicted transiently acting elements in the promoter region (Fig. 4D). The results showed abundant regulatory elements in the promoter region, mainly those for the phytohormones gibberellin, abscisic acid, and auxin; as well as responses to abiotic, anaerobic, low-temperature, drought, hypoxia, and light stresses; and biological growth processes including circadian rhythm, seeds, zein metabolism, meristem expression, endosperm expression, and flavonoid biosynthesis. These results indicate that members of the *G. barbadense* PIN gene family are involved in plant growth, differentiation, regulation, and responses to various biotic and abiotic stresses.

## PIN family gene collinearity analysis

Comprehensive genome collinearity analysis results for the four cultivated cotton species (*G. barbadense, G. hirsutum, G. raimondii,* and *G. arboreum*) are shown in Fig. 5. PIN gene family members had 22 sets of collinearity in *G. barbadense* and *G. hirsutum*, three sets of collinearity in *G. raimondii*, and no collinearity in *G. arboreum*. These results indicate that each member of this gene family in *G. arboreum* remains highly independent in diploid cotton material; after combining to form tetraploid cotton, they corresponded to each other on chromosomes A and D, but then were lost, possibly due to duplication of function. There were 63, 42, and 22 sets of collinear relationships between *G. barbadense* and *G. hirsutum*, *G. raimondii*, and *G. arboreum*, respectively. These results may simply correspond to numbers of genes; however, the lack of collinearity between *G. barbadense* and *G. arboreum* may indicate that *G. arboreum* PIN family genes were largely lost through recombination into the *G. barbadense* gene.

## Expression profiling and endogenous auxin correlation analysis of PIN family genes

We analyzed the FPKM values of the PIN family genes in transcriptome data from Hai7124, TM-1, PimaS-7, and 5917, and found that approximately 40% of PIN family

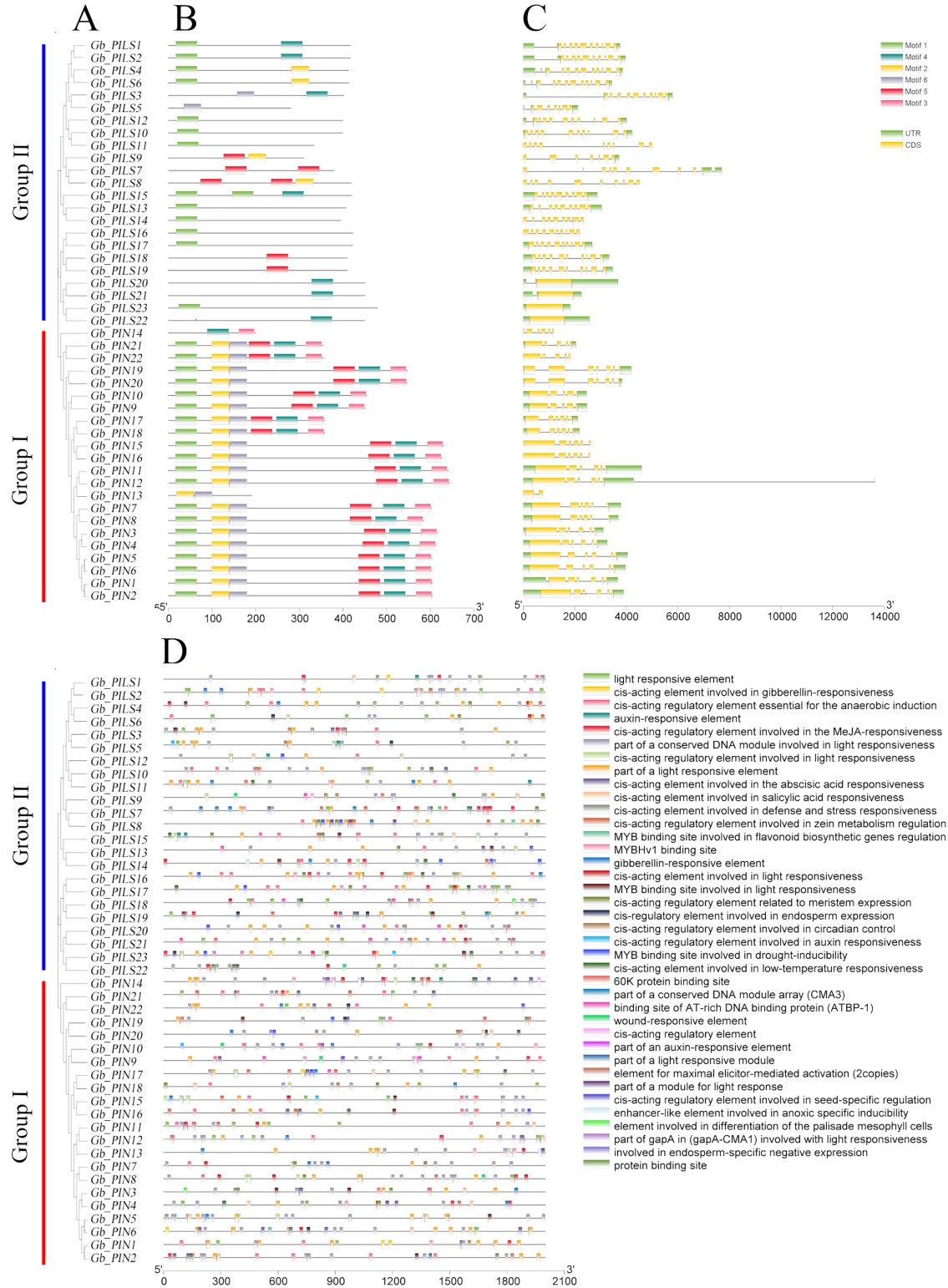

**Figure 4 Analysis of the gene structures and promoter structures of PIN family genes in *G. barbadense*.** (A) GbPIN gene evolutionary tree, (B) motif, (C) structure, and (D) promoter cis-acting elements.

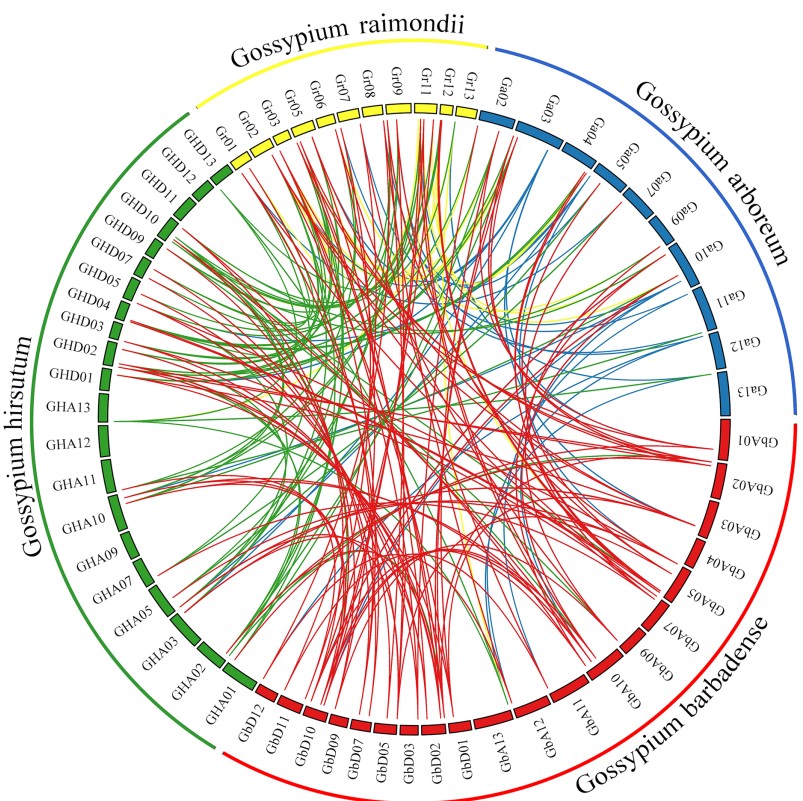

**Figure 5 Genome collinearity map of PIN family genes in the four major cultivated cotton species examined in this study.**

genes were not detected in these four materials. Analysis of the Hai7124 and TM-1 transcriptomes revealed that 21 PIN genes had significant tissue-specific expression, and multiple genes were differentially expressed at different stages of fiber development. The expression levels of *GBPIN1/GHPIN1*, *GBPIN2/GBPIN2*, *GBPILS1/GHPILS1* and 10 other genes were significantly higher in roots and stems than in leaves, and those of 11 genes including *GBPIN11/GHPIN11*, *GBPIN12/GHPIN12*, and *GBPILS22/GHPILS22* were significantly higher in stems and leaves than in roots; these genes may be involved in regulating the local accumulation of auxin in plant organs. Six genes including *GBPILS18/GHPILS18/GbPILS18* and *GBPILS19/GHPILS19/GbPILS19* were expressed at 0–35 DPA, during fiber development, and the overall expression trend showed slightly higher expression at 0–35 DPA (Figs. 6A–6D).

Endogenous auxin content in cotton fibers was significantly lower in PimaS-7 than in 5917 at 10, 15, and 20 DPA, and that in 5917 was significantly lower at 15 and 20 DPA than at 10 and 30 DPA, respectively. In both materials, auxin content first increased, then decreased, and then increased again. In PimaS-7, auxin content peaked at 5 and 30 DPA, whereas that of 5917 peaked at 10 and 30 DPA; auxin content was significantly higher in 5917 than in PimaS-7 (Fig. 7A).

Correlation analysis showed that the expression levels of only five genes were significantly correlated with endogenous auxin content in PimaS-7 and 5917 at different

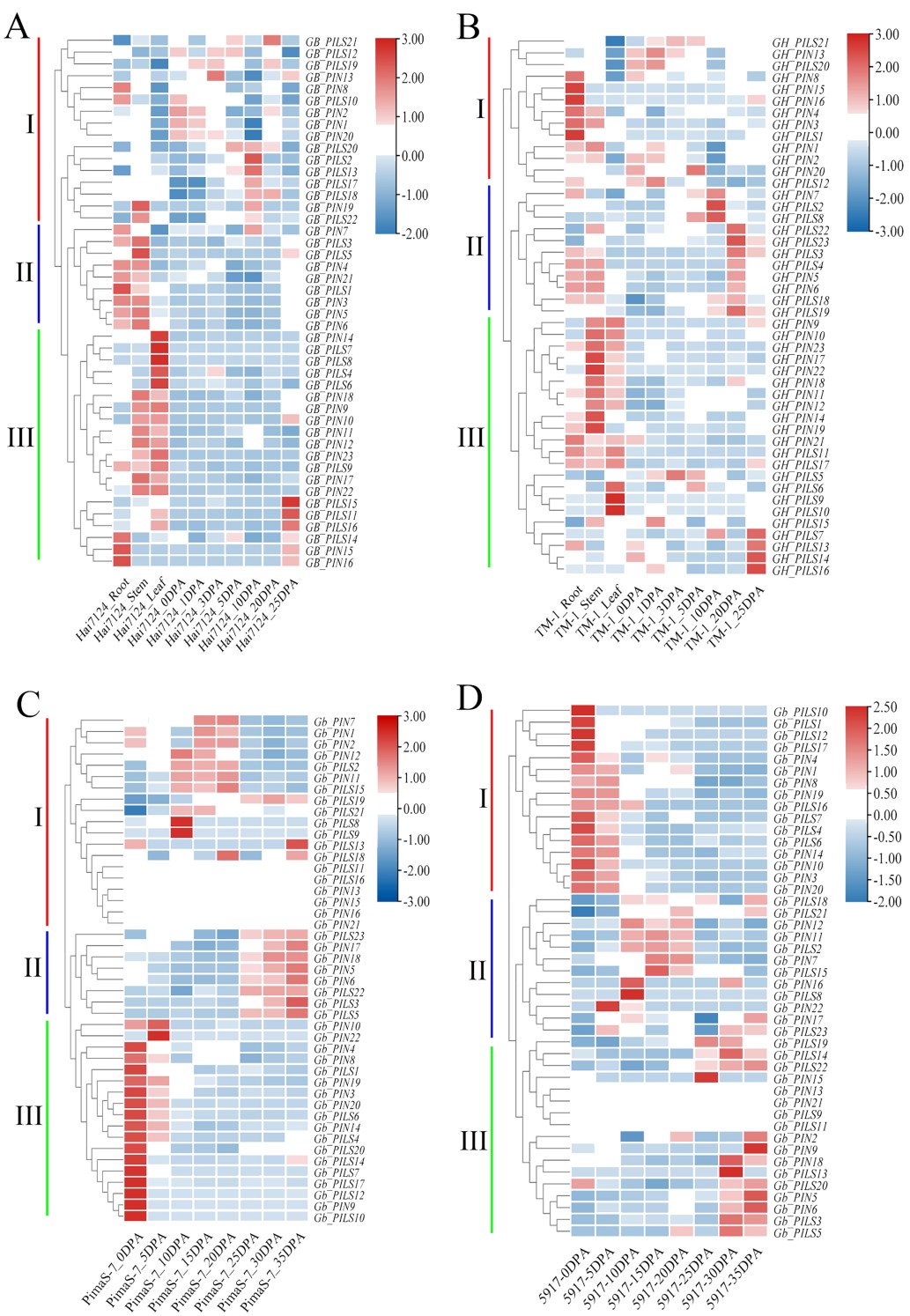

**Figure 6 Heat map of PIN family gene expression.** (A) Hai7124, (B) TM-1, (C) PimaS-7, and (D) 5917 transcriptome materials during rhizome leaf fiber development from 0 to 35 days post-anthesis (DPA).

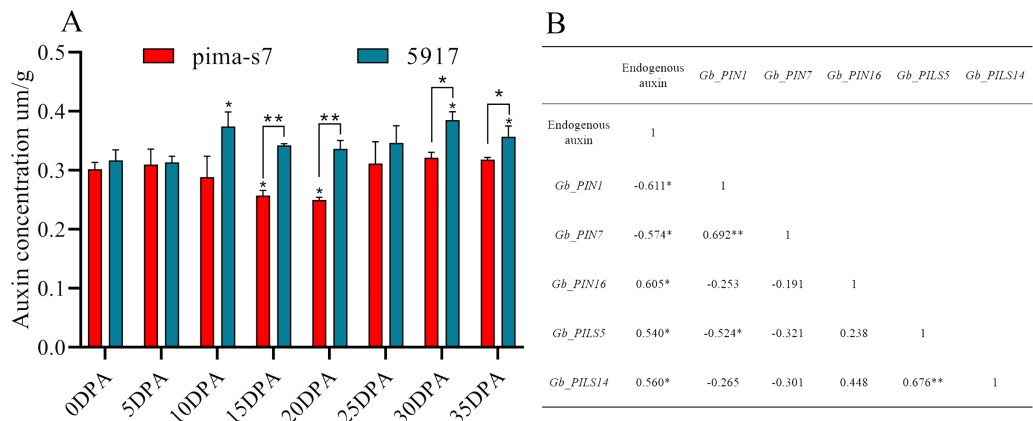

**Figure 7 Endogenous auxin content determination and correlation between auxin levels and fragments per kilobase of transcripts per million mapped reads (FPKM) in PIN family genes.** (A) Endogenous auxin content in PimaS-7 and 5917 fiber samples (0–35 DPA). (B) Correlation between endogenous auxin content and FPKM values of corresponding PIN family genes in PimaS-7 and 5917 fiber samples from 0 to 35 DPA. $^*P < 0.05$, $^{**}P < 0.01$. 

stages of fiber development. *GbPIN1* and *GbPIN7* were negatively correlated with auxin content, whereas *GbPIN16*, *GbPILS5*, *GbPILS14* were positively correlated (Fig. 7). However, as shown in Fig. 6, the expression levels of five genes (*GbPIN1*, *GbPIN16*, *GbPILS5*, and *GbPILS14*) were very low at various stages of fiber development; only *GbPIN7* had a high expression level, which differed between the two materials and was directly related to endogenous auxin content.

## qRT-PCR analysis of GbPIN family genes

Further analysis of the PIN I subgroup showed that 8 of 10 genes in this subgroup had detectable transcripts in the transcriptome data. Therefore, qRT-PCR analysis was performed on these eight genes; none of them showed consistent patterns in the fibrous tissues from 5 to 30 DPA, and some differences were observed between the two materials (Fig. 8). Three genes (*GbPIN3*, *GbPIN4*, and *GbPIN8*) showed a trend of increasing expression from 5 to 25 DPA, followed by low expression (Figs. 8A, 8D, 8H).
The expression levels of four genes (*GbPIN1*, *GbPIN2*, *GbPIN5*, and *GbPIN6*) increased at 5–15 and 20 DPA, and decreased thereafter in 5917, whereas those of PimaS-7 gradually increased from 5 to 25 DPA, and then decreased (Figs. 8A, 8B, 8E, 8F). Thus, the expression of these four genes peaked later in the low-strength fiber material than in the high fiber-strength material, suggesting that these genes influence secondary wall thickening. The expression trend of the *GbPIN7* gene was relatively consistent between the two materials, gradually increasing until 20 DPA, and then slowly decreasing (Fig. 8G).

## Functional verification of the GbPin7 gene

Because *GbPIN7* gene expression was observed in both the transcriptome and fluorescence quantitative analysis results and significant differences were detected between the two research materials, we conducted further transient silencing analysis of the *GbPIN7* gene. The leaves of positive plants change from green to white after 15 days (Figs. 9A and 9B).

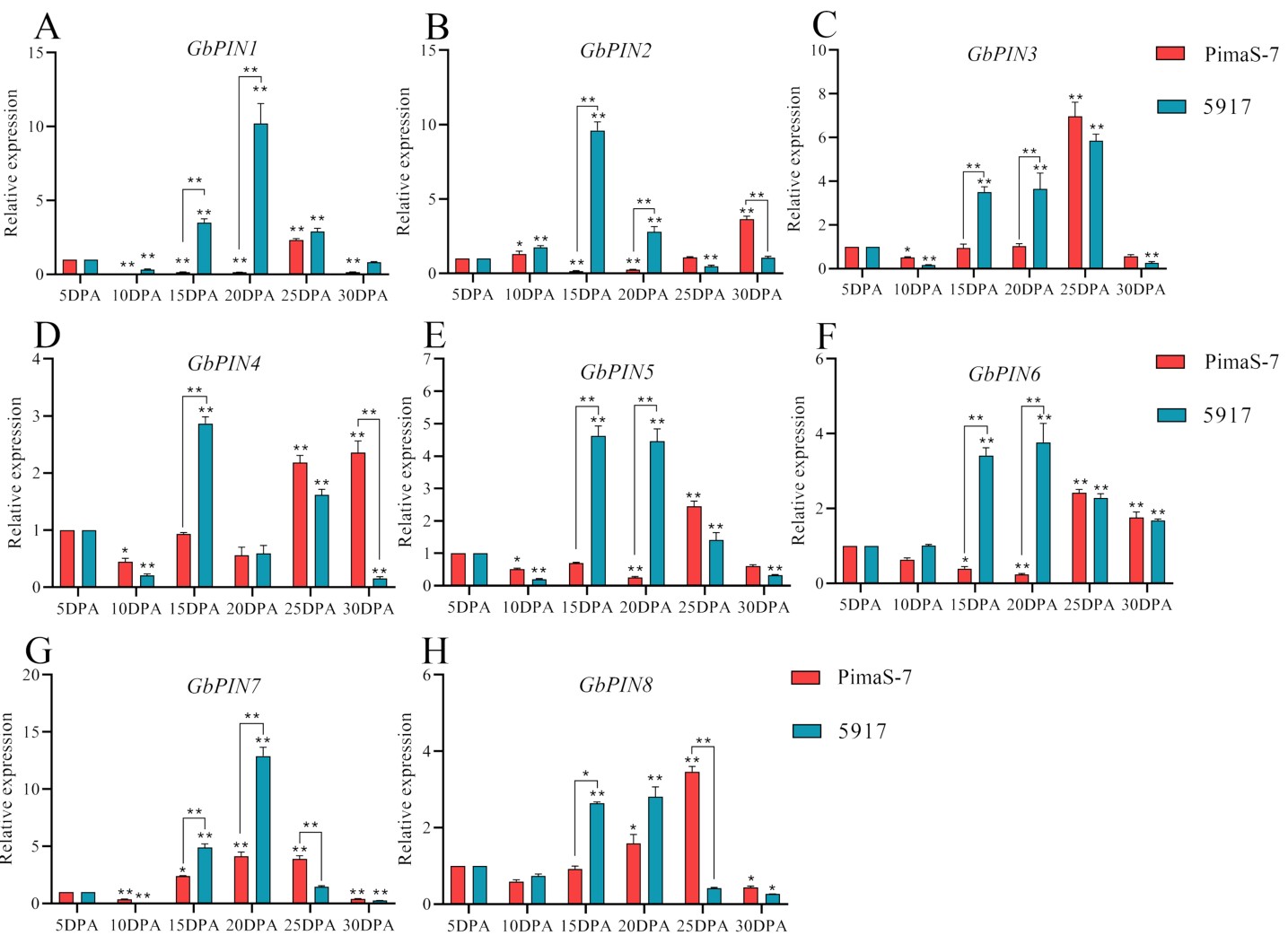

**Figure 8 Quantitative polymerase chain reaction (qPCR) analysis of the GbPIN1 subfamily.** (A–H) PIN1 subfamily expression during fiber development from 5 to 30 DPA expression. Asterisks indicate significant differences between time periods. $^*P < 0.05$, $^{**}P < 0.01$.

Silencing efficiency was confirmed in the experimental and negative control groups using qPCR (Fig. 9C). Following silencing, the plants were cultured for 45 days under the same water and fertilizer conditions. Plant growth was significantly accelerated in both PimaS-7 and 5917 materials (Fig. 9D). Endogenous auxin content was measured in the upper leaves and stems of the same plants; the results showed significantly reduced auxin content.

## DISCUSSION

Two previous studies (*Zhang et al., 2017a*; *Li et al., 2017a*) identified 17 PIN family genes in *G. hirsutum*, whereas in this study, we identified 45, 46, 24, and 24 PIN family genes in *G. barbadense*, *G. hirsutum*, *G. arboreum*, and *G. raimondii*, respectively. These differences between studies may be the result of different reference genomes used; with the continuous development of sequencing technology, our knowledge of the cotton genome is improving, and PIN family gene identification results are increasingly reliable. The present study

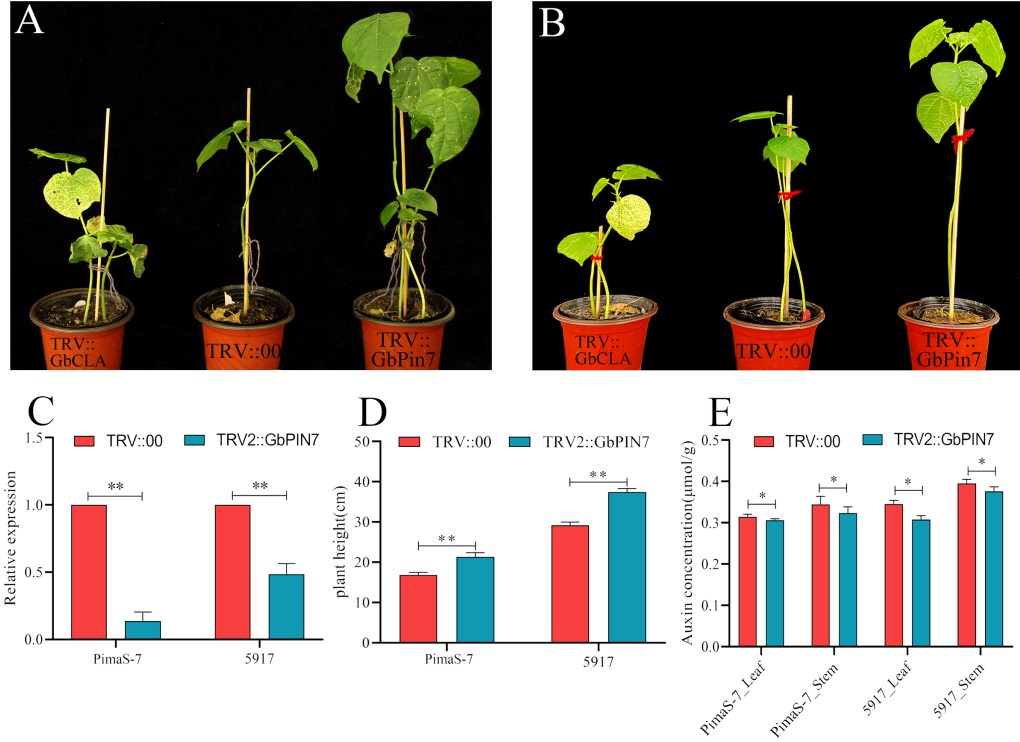

**Figure 9 Diagram of GbPIN7 gene virus-induced gene silencing (VIGS) analysis results.** (A, B) Comparison of PimaS-7 and 5917 cotton plants after 10 days of infection with Agrobacterium containing TRV strains. (C) Silencing efficiency detected by qRT-PCR. (D) Plant height of cotton plants 45 days after infection. (E) Determination of endogenous auxin content in shoot tips and leaves of cotton plants 45 days after infestation.

comprehensively identified all sequences with the conserved structure of Mem_trans, whereas previous studies identified members of the *G. hirsutum* PIN gene family through sequence alignment of *Arabidopsis* PIN family genes. Therefore, our findings extend previous results to quantitatively identify additional members of the PIN gene family.

In terms of the number of PIN gene family members, the allotetraploid cotton (*G. barbadense* and *G. hirsutum*) exhibited changes in the number of PIN genes compared with their diploid ancestors (*G. arboreum* and *G. raimondii*). The PIN5/6/8 and PILS5 subgroups of *G. barbadense* had two fewer genes than the sum of genes in Raymond's cotton and Asian cotton, which indicates that some genes in this subgroup may not be necessary for cotton development. The number of genes in *PILS1/3* subgroup increased during evolution. Compared to the original species, the number of genes in this subgroup increased by one, suggesting that the functions of this subgroup of genes functionally expanded during evolution. The remaining four subgroup genes were well represented in terms of quantity and the sequence relationship during evolution from the original cotton material to allotetraploid cotton, indicating that members of this gene family are involved in cotton growth and development.

Multiple members of the PIN gene family are involved in auxin transport between cells in plant tissues such as roots, stems, and cotton fibers. In *Arabidopsis*, *AtPIN1*, *AtPIN2*,

*AtPIN3*, *AtPIN4*, and *AtPIN7* are involved in intercellular auxin transport in roots (*Křeček et al., 2009*). *AtPIN1* and *AtPIN3* are highly expressed in roots, determining the size of the main root meristem, and they determine the taproot growth rate (*Omelyanchuk et al., 2016*). In this study, 10 genes including *GBPIN1*, *GBPIN2*, and *GBPIN3* were significantly highly expressed in roots and stems, but barely expressed in leaves. This finding is highly consistent with those of previous studies. PIN family genes showed significant tissue-specific expression, indicating that these genes play important roles in root development and local auxin accumulation of auxin in cotton.

In *G. hirsutum*, transcripts of the PIN-homologous gene *GhPIN3a* were detected in the outer ovule coat (including fibroblasts) in addition to the nucleolus (*Zhang et al., 2017b*; *Zeng et al., 2019*), suggesting that GhPIN3a is involved in auxin regulation in the ovule epidermis. Gene chip analysis showed similar localization results of the AtPIN3 gene in *Arabidopsis* during ovule development (*Le et al., 2010*). Previous studies have reported that the further differentiation of ovule epidermal cells requires the participation of *PIN3*, which led to prediction of the potential auxin flow path in the cotton ovule: auxin moves from the ovule root through the vascular bundle to the fiber cells and nucleolus (*Guan et al., 2011*). In this study, transcriptomes of TM-1 and Hai7124 materials showed that both PIN11 and PIN12 genes were expressed in rhizomes and leaves, and expression levels decreased in the order stem > leaf > root. During fiber development, these genes were highly expressed in fiber and ovule cells at 10 DPA, and in fiber cells of both PimaS-7 and 5917 at 10, 15, and 20 DPA. Previous studies have reported that PIN11 and PIN12 play crucial roles in cotton plant growth and fiber cell initiation, and may also play an important role in regulating the growth of fiber cells.

*Bai et al. (2008)* measured endogenous auxin content at various stages of cotton fiber development, and found it changed over time; they constructed expression profiles for different tissues, organs, and developmental stages for each PIN family gene (*Forestan, Farinati & Varotto, 2012*; *Yang et al., 2019*). However, no previous studies have examined the correlation between PIN gene expression and endogenous auxin content during fiber development, as conducted in the present study. Our results indicate that multiple PIN family genes are directly correlated with endogenous auxin content during cotton fiber development.

The *PIN7* gene can influence plant root tillering (*Carraro et al., 2006*), negative phototrophic root growth (*Xu et al., 2014*), epidermal hair development (*Zhang et al., 2017a*), tissue differentiation (*Oliveros-Valenzuela et al., 2007*), and flower development (*Gälweiler et al., 1998*), but not plant development. The reason for this remains unclear. In this study, transient VIGS analysis showed that this gene regulated the accumulation of endogenous auxin in plant tissues. This finding may explain how *PIN7* regulates the growth of various plant tissues.

## CONCLUSION

This study comprehensively analyzed the PIN gene family in sea island cotton. The results showed that multiple genes in this family were directly related to the accumulation of endogenous auxin during cotton fiber development, thereby regulating cotton fiber

development and silencing the *GbPIN7* gene. The gene downregulation affected plant growth promotion. This study provides a reliable theoretical basis for studying the PIN auxin transporter in sea island cotton and will further assist the molecular breeding process in cotton.

### Funding

The research was supported by the Hainan Provincial Joint Project of Sanya Yazhou Bay Science and Technology City (Grant No. 2021JJLH0064), China Postdoctoral Science Foundation (Grant No. 2020M683711XB), and Hainan Yazhou Bay Seed Laboratory (Grant No. B21Y10204). The funders had no role in study design, data collection and analysis, decision to publish, or preparation of the manuscript.

### Grant Disclosures

The following grant information was disclosed by the authors:
Hainan Provincial Joint Project of Sanya Yazhou Bay Science and Technology City: 2021JJLH0064.
China Postdoctoral Science Foundation: 2020M683711XB.
Hainan Yazhou Bay Seed Laboratory: B21Y10204.

### Competing Interests

The authors declare that they have no competing interests.

### Author Contributions

- Yilei Long conceived and designed the experiments, performed the experiments, analyzed the data, prepared figures and/or tables, authored or reviewed drafts of the article, and approved the final draft.
- Quanjia Chen conceived and designed the experiments, prepared figures and/or tables, authored or reviewed drafts of the article, and approved the final draft.
- Yanying Qu conceived and designed the experiments, prepared figures and/or tables, and approved the final draft.
- Pengfei Liu performed the experiments, authored or reviewed drafts of the article, and approved the final draft.
- Yang Jiao performed the experiments, authored or reviewed drafts of the article, and approved the final draft.
- Yongsheng Cai analyzed the data, authored or reviewed drafts of the article, and approved the final draft.
- Xiaojuan Deng analyzed the data, authored or reviewed drafts of the article, and approved the final draft.
- Kai Zheng conceived and designed the experiments, authored or reviewed drafts of the article, and approved the final draft.

## DNA Deposition

The following information was supplied regarding the deposition of DNA sequences:

The RNA sequence raw data is available at NCBI, TM-1 and Hai7124: PRJNA450479. PimaS-7 and 5917 is available at NCBI: PRJNA741534.

## Data Availability

The DNA seq, protein seq and Genome annotation files are available at CottonGen:

– *G. hirsutum* TM-1 (ZJU-AD1_v2.1_a1.0)

https://www.cottongen.org/species/Gossypium_hirsutum/ZJU-AD1_v2.1

– *G. barbadense* Pima3–79 (HAU_v2.0)

https://www.cottongen.org/species/Gossypium_barbadense/HAU-AD2_genome_v2.0

– *Gossypium arboreum* (CRI-A2_v1.0)

https://www.cottongen.org/species/Gossypium_arboreum/CRI-A2_genome_v1.0

– *Gossypium raimondii* (JGI_221_v2.1)

https://www.cottongen.org/species/Gossypium_raimondii/jgi_genome_221

## Supplemental Information

Supplemental information for this article can be found online at http://dx.doi.org/10.7717/peerj.14236#supplemental-information.

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
