# Peer review of "Identification and functional analysis of PIN family genes in Gossypium barbadense"

_PeerJ, doi:10.7717/peerj.14236_

## Round 0.1 · original submission · Major Revisions

Dear Authors,

Thank you for submitting the Manuscript to PeerJ. We have received comments from Reviewers.

Please reply to the Reviewer's comments, so that we can check it further.

Thanks

Reviewer 1 ·

Basic reporting

no comment

Experimental design

no comment

Validity of the findings

no comment

Additional comments

The study did genome-wide identification of the PIN family genes in four cotton species, with a focus on those in Gossypium barbadense. On top of analyses of the basic features of the PIN genes and the corresponding proteins, temporal transcriptomic data were generated from two G. barbadense accessions with significant difference in fibre strength and the auxin content in developing fibres from the corresponding stages (I assume that’s the case) were measured. The correlation between the expression level of PIN encoding genes and the auxin content in developing fibre was investigated.
Given the significantly different fibre strength of the two G. barbadense accessions used in the study and the importance of PIN genes in plant development, the topic of the study is of interest; however, the manuscript has several major issues that have to be addressed before it can be accepted for publication.
1. The aim of the study seems to be identification and functional analysis of the PIN genes (based on the title). While, based on the fibre transcriptomic analysis results, GbPIN7 was selected for VIGS in the two G. barbadense accessions, it’s unclear the impact of down-regulation of the gene on fibre development, despite the observation of enhanced plant growth. So the scientific question on the role of PIN genes in fibre development was not properly addressed.
2. The methods used in identification and characterization of PIN genes were not clear at all. There is no information about the samples used in measurement of endogenous auxin content, and the method used in the measurement was not described with sufficient detail. No information was available for the methods used in transcriptome analysis.
3. Many experimental data and results were not properly interpreted or even mis-interpretated. For instance, “the lack of collinearity between G. barbadense and G. arboreum may indicate that G. arboreum PIN family genes were largely lost through recombination into the G. barbadense gene.” There are too many such examples to be pointed out one by one.
4. Data and results were poorly presented. Sometimes, it’s hard to follow what exactly the authors have tried to present.
5. The major conclusions (for instance, “our results help clarify the roles of PIN family genes in cotton fiber development”) were not supported by the data presented.
6. Overall, the manuscript requires overhaul in many aspects, including but not limited to English language, data interpretation and presentation, and drawing conclusions.

·

Basic reporting

The comments are specified in 'Additional comments'.

Experimental design

The comments are specified in 'Additional comments'.

Validity of the findings

The comments are specified in 'Additional comments'.

Additional comments

The authors identified and analyzed a total of 138 PIN family genes in four cotton species G. barbadense, G. hirsutum, G. raimondii, and G. arboretum. They assorted the PIN genes into seven categories. They found that GbPIN gene family members were widely distributed on 20 chromosomes, and most had repeated duplication events. They found that some genes had differential expression patterns in different stages of fiber development through transcriptome analysis. The qRT-PCR and VIGS were also used to verify the PIN gene expression characteristics in different stages of fiber development. This study provides comprehensive analyses of the expression characteristics and roles of PIN family genes in G. barbadense during fiber development. The manuscript has merits for a publication in PeerJ, but it still has some problems that need the authors to consider revising.

Lines 36-38, as different reports refer to different roles, therefore, better specify each role with its reference instead of put the references in a whole at the end of the statement of roles. This may make readers easier to tract the references.

Lines 82-83, the statement ‘QTL…’ needs reference to support.

Lines 175-203, sections ‘Physicochemical properties of PIN genes’ and ‘Gene evolution and chromosomal location analysis results’ should be combined into one and the content should be reorganized. Delete the duplicated descriptions.

Figure 5, labels of G. raimondii and G. arboreum are not correct.

Lines 230-240, conclusion of collinearity analysis is not correct. you identified 45 PIN in Gb, 46 in Gh, 24 in Gr and 24 in Ga. In figure 5, it can be observed collinearity between Gb and Ga. Then, how can you drew the result ‘that G. arboreum PIN family genes were largely lost through recombination into the G. barbadense gene’?

Lines 270-271, is your statement correct?

Lines 287-288, the meaning of the sentence is not clear.

Reviewer 3 ·

Basic reporting

The present study reports the role of PIN proteins that play an important regulatory role in plant growth and development in Gossypium barbadense. Overall the research article has been written well and adds interesting insights to existing literature. The language of the article is good from a reader point of view.
The authors have described the literature well, however interesting they forget to cite or mention outcome of research finding of He et al.2017 (BMC Genomics) who completed a similar study on PIN proteins in cotton. I r recommend authors to add more relevant literature to make the study more interesting.

Experimental design

Experimental design of the study has been satisfactory, all methodologies have been followed as per standard. Research question has been defined well.

Validity of the findings

The present comprises of in silico study and wet lab validations. A total of 138 PIN family genes were identified in the four cotton species; the genes were divided into seven subgroups. Overall the study provided comprehensive analyses of the roles of PIN family genes in G. barbadense and their expression during cotton fiber development. The results are really helpful for a plant breeder and biotechnologist to translate this information in some defined traits of fibre improvement in cotton.

Additional comments

Overall the article is well composed, needs some linkage with current literature by adding relevant citations.

---

## Round 0.2 · Minor Revisions

Dear Authors,
Thanks for submitting the revised version of the manuscript.
One of the reviewers has sent addiitonal comments .
Please address the minor comments

·

Basic reporting

Please see the additional comments.

Experimental design

Please see the additional comments.

Validity of the findings

Please see the additional comments.

Additional comments

Authors have greatly improved the manuscript in current version. But there are still places that require additional revisions.
Lines 26-28, The sentence is not clear.
Lines 37-40, punctuations including comma and brackets, need correction. The author need to go through the whole manuscript carefully to find the same mistakes and correct them.
Lines 183-184, punctuations are not correct. The sentence is not complete and its meaning is not clear.
191-194, It is not clear the authors describe.
English language is poor, needs polishing. Suggest they hire a language editing company to polish the English language.

---

## Round 0.3 · accepted · Accept

The authors have addressed the minor comments and the manuscript in its current form is acceptable now.